# Incidence of Fat Embolism Syndrome in Femur Fractures and Its Associated Risk Factors over Time—A Systematic Review

**DOI:** 10.3390/jcm10122733

**Published:** 2021-06-21

**Authors:** Maximilian Lempert, Sascha Halvachizadeh, Prasad Ellanti, Roman Pfeifer, Jakob Hax, Kai O. Jensen, Hans-Christoph Pape

**Affiliations:** 1Department of Trauma, University Hospital Zurich, Raemistr. 100, 8091 Zürich, Switzerland; Sascha.Halvachizadeh@usz.ch (S.H.); Roman.Pfeifer@usz.ch (R.P.); Jakob.hax@usz.ch (J.H.); kaioliver.jensen@usz.ch (K.O.J.); Hans-Christoph.Pape@usz.ch (H.-C.P.); 2Department of Trauma and Orthopedics, St. James’s Hospital, Dublin-8, Ireland; prasad.ellanti@gmail.com

**Keywords:** fat embolism syndrome, trauma, long bone fractures, femur fracture

## Abstract

Background: Fat embolism (FE) continues to be mentioned as a substantial complication following acute femur fractures. The aim of this systematic review was to test the hypotheses that the incidence of fat embolism syndrome (FES) has decreased since its description and that specific injury patterns predispose to its development. Materials and Methods: Data Sources: MEDLINE, Embase, PubMed, and Cochrane Central Register of Controlled Trials databases were searched for articles from 1 January 1960 to 31 December 2019. Study Selection: Original articles that provide information on the rate of FES, associated femoral injury patterns, and therapeutic and diagnostic recommendations were included. Data Extraction: Two authors independently extracted data using a predesigned form. Statistics: Three different periods were separated based on the diagnostic and treatment changes: Group 1: 1 January 1960–12 December 1979, Group 2: 1 January 1980–1 December 1999, and Group 3: 1 January 2000–31 December 2019, chi-square test, χ^2^ test for group comparisons of categorical variables, *p*-value < 0.05. Results: Fifteen articles were included (*n* = 3095 patients). The incidence of FES decreased over time (Group 1: 7.9%, Group 2: 4.8%, and Group 3: 1.7% (*p* < 0.001)). FES rate according to injury pattern: unilateral high-energy fractures (2.9%) had a significantly lower FES rate than pathological fractures (3.3%) and bilateral high-energy fractures (4.6%) (*p* < 0.001). Conclusions: There has been a significant decrease in the incidence of FES over time. The injury pattern impacts the frequency of FES. The diagnostic and therapeutic approach to FES remains highly heterogenic to this day.

## 1. Introduction

Fat embolism continues to represent a major complication in patients with acute trauma [1,2]. According to the current understanding, a subacute multisystem dysfunction can develop, typically presenting with a bimodal onset of nonspecific symptoms such as tachycardia, tachypnoea, and fever sometimes days after the initial trauma [3]. Subsequently, the fully developed syndrome can include pulmonary failure, neurologic symptoms, and the possible development of multiple organ failure [4,5]. The symptoms following acute fat embolization may vary between subclinical changes (e.g., evidence of fat globules in retinal arterioles) and fulminant acute lung failure with subsequent organ dysfunction [6].

Although Lower et al. provided the first description of FES about 300 years ago in dogs, the current clinical relevance for patients with long bone fractures was only recognized in the 1970s. Back then, FES was a major reason to delay surgeries of long bone fractures [1,7]. FES was associated with simple techniques of artificial ventilation. It was therefore unclear whether early FES or artificial ventilation were among the main reasons for pulmonary complications after fracture fixation. It was recognized that high-pressure changes during ventilatory support were among the causes of endothelial disruption, pneumothorax, and ARDS [5]. Likewise, the first diagnostic approaches to differentiate between ventilator-associated complications and FES relied on the direct proof of increased fat contents in the systemic circulation, which may be a rather transient phenomenon [8].

These diagnostic criteria were modified later by the inclusion of alternative indexes, and other scoring systems have been proposed [9,10,11], none of which have been validated prospectively. The most widely accepted diagnostic criteria were introduced by Gurd et al. in 1972 (Appendix A) [12]. Subsequently, computed tomography and MRI techniques have been added and have replaced the direct proof of fat in the vascular system [13,14,15]. These changes in diagnostic options might explain the fact that the reported incidences of FE are highly variable (47–100%) [12,16,17,18]. Autopsy reports are of considerable significance, and signs of FE are seen in 68–82% [19,20] of cases after major trauma. Acute embolization of fat during reaming has been detected in 88% of patients with the use of intraoperative transesophageal echocardiography [21]. Likewise, both the incidence of FES ranging from 0.8% to 23% [22,23] and the mortality rate ranging from 1% to 10% [22,24,25,26] are highly variable in the literature.

As of today, even though it continues to be a major complication of orthopedic surgeries, there appears to be uncertainty regarding the trend in the incidence of FES over time, the relationship to injury patterns, and the current clinical recommendations.

Therefore, we performed a systematic review on fat embolism syndrome and aimed to answer the following questions:Are the diagnostic standards applicable to all trauma patients?Has the incidence of FES changed over time since its clinical description?Does the injury pattern (unilateral versus bilateral acute high-energy fractures versus pathological fractures) have an impact on the incidence of FES?Are specific evidence-based therapeutic recommendations available in the recent literature?

## 2. Methods

This study was conducted according to the Preferred Reporting Items for Systematic Reviews and Meta-analyses (PRISMA) guidelines (Figure 1) [27]. The study was registered with PROSPERO (registration number: CRD42021238862).

### 2.1. Search Strategy

The systematic literature search included the MEDLINE, PubMed, EMBASE, Web of Science, and Cochrane Library databases for articles in the English or German language if published between 1 January 1960, and 1 June 2019. The following MeSH terms were used: fat embolism syndrome; fat embolism, and femoral fracture; pathologic femoral fracture; and bilateral femoral fracture. Additionally, the reference lists of the selected studies and related systematic reviews were screened to identify any relevant studies we missed in our electronic search, as they might not have been available in the databases we used.

### 2.2. Study Selection

Two authors, both experienced in systematic reviews (ML and JH), independently selected the studies. Discrepancies were solved by consensus and arbitration with another experienced author (HCP) if needed. Studies were first screened by title and/or abstract. For studies included after the screening, we obtained full texts for formal inclusion into the current analysis.

### 2.3. Inclusion and Exclusion Criteria

We included randomized trials and observational studies published between 1 January 1960 and 31 December 2019 in the English or German language. Data sources included the MEDLINE, Embase, Web of Science, and Cochrane Register of Controlled Trial database (CENTRAL) regional databases and references of the included studies.

In general, trials that reported about isolated major fractures and multiple injured patients and their incidences of complications were included. Further specific inclusion criteria were used as follows:isolated and multiple femur fractures caused by high- or low-energy injuries and pathological fractures;description of primary and secondary outcomes (mortality, complications, length of intensive care unit stay, or length of hospital stay);original patient data from a single- or multicenter approach.

Injury pattern and trauma mechanism were stratified into unilateral femoral fractures (unilateral), bilateral femoral fractures (bilateral), and unilateral pathological femoral fractures (pathological).

Further, the diagnostic criteria of FES and adherence to clinical diagnostic recommendations, as well as mortality rate and time to surgical treatment, were noted.

One author (JH) assessed the quality of randomized controlled trials (RCTs) using the “Cochrane Collaboration’s tool for assessing the risk of bias” (Table 1), which was integrated into the selection procedures [28].

We performed a risk of bias analysis using the Cochrane Risk of Bias Assessment tool [28]. The results of five nonrandomized studies [23,25,29,30,31] are shown in Table 1. Of the 15 studies analyzed, 9 were nonrandomized and noncomparative studies [22,26,32,33,34,35,36,37,38]. In the literature, no criteria are defined to assess the risk of bias for noncomparative studies. One of the 15 trials was a randomized comparative study. It was analyzed with the Cochrane Collaboration’s tool for assessing the risk of bias in randomized trials and rated with a high risk [39].

### 2.4. Group Distribution

All articles were distributed according to the time of publication.

Three different 20-year periods were selected in order to simplify the statistics and to give an adequate overview:Group 1 (1 January 1960–31 December 1979),Group 2 (1 January 1980–31 December 1999),Group 3 (1 January 2000–31 December 2019).

For publications with overlapping study periods, the study was stratified according to its main study period within these three timeframes.

### 2.5. Statistics

Descriptive statistics included the presentation of continuous variables as the mean and standard deviation and of categorical variables as the numbers and percentages. Group comparisons of categorical variables were either performed with Fisher’s Exact test or the chi-square test as appropriate. In the cases of group comparisons with less than five samples, a Fisher’s Exact test was utilised. Group comparisons of the categorical variables were either performed with a Student’s *t*-test or ANOVA, as appropriate.

Statistical analyses and graphical displays were performed using GraphPad Prism (version 8.0.0 for Windows, GraphPad Software, San Diego, CA, USA, (www.graphpad.com)). A *p*-value of <0.05 was considered statistically significant.

## 3. Results

The initial search yielded a total of 1024 publications, of which 890 were unique after screening for titles and abstracts, and 85 articles were selected for full-text screening. Fifteen articles met the inclusion criteria and were available for the critical appraisal procedure (Figure 1).

### 3.1. Study Populations

All the demographic data and injury characteristics are summarized in Table 2. These do not explain any of the differences in the incidences and complications listed below.

### 3.2. Diagnostic Criteria

The diagnostic tools used by different study groups are depicted in Table 3. The majority (8/15) applied Gurd’s criteria [12], while the others used “clinical evidence”, relied on the results of autopsy reports, or utilized indirect criteria, such as the development of “acute cardiorespiratory and vascular dysfunction”.

### 3.3. Changes in the Incidence of FES over Time

The incidence of FES decreased stepwise from 7.93% (*p* < 0.01) to 1.69% (*p* < 0.01) when the three different time periods were compared. This decrease was more sustained in the last decade when compared with the previous ones (Figure 2).

### 3.4. FES and Injury Patterns

The highest incidence of FES was observed in bilateral femoral fractures. Furthermore, the pathological unilateral fractures showed a significantly higher incidence of FES compared to the isolated unilateral fractures (unilateral high-energy fractures (2.97%; *p* < 0.01), pathological fractures (3.34%; *p* < 0.01), and bilateral fractures (4.65%; *p* < 0.01) (Figure 3).

### 3.5. Treatment Recommendations

Structured recommendations for the treatment of FES were not available in the recent literature (group 3) or in the more-distant past. Although some publications focused on the avoidance of FES complications, a distinct treatment recommendation was not available, and the recommendations for pulmonary, cerebral, and other forms of FES were variable. This included the specific criteria for admission to an intensive care unit (ICU). Artificial ventilation represented the only common variant represented in all publications (data not shown).

## 4. Discussion

Fat embolism syndrome (FES) continues to be mentioned in the current literature as a potentially devastating complication following orthopedic trauma involving long-bone fractures. Although the triad of hypoxia, petechial rash, and altered mental status appears to be less common, the concern among surgeons continues to be relevant [40].

This applies especially to intraoperative fat embolism, which may be associated with rapid postoperative organ dysfunction or even a lethal outcome [41,42].

Our manuscript appears to be the first to compare the diagnostic principles, variations in the incidences of FES over time, and associations with injury patterns. The findings of our literature search were clear-cut for all four questions addressed initially as follows:Regarding diagnostic standards, most authors appeared to mainly rely on the Gurd’s criteria to diagnose the clinical syndrome. Magnetic resonance imaging (T2) has been routinely used as an adjunct tool in the radiologic literature [43], but the studies looked over in our review did not routinely use it.The incidence of FES has decreased from almost 8% to close to 2% since its clinical description. Our results indicated that this appears to apply especially for the period after the year 2000 (*p* < 0.01).We found a clear association between the incidence of FES and the injury pattern as follows: unilateral fractures (traumatic and pathological) were associated with a lower incidence of FES than bilateral fractures. Additionally, unilateral pathological fractures were associated with a higher incidence of FES compared to traumatic unilateral fractures.No universal recommendations in the literature regarding therapeutic strategies were found. Some authors recommended using steroids in the early years [9], but this treatment was rapidly discontinued due to infectious complications. Some authors even recommended prophylactic positioning of these particular patients at risk in the prone position [44].

In summary, except for the requirement of ventilatory support—including the use of ECMO [45]—no universal treatment was recommended.

Regarding our first question, the wide range of publications may have played a role. Since the introduction of Gurd’s criteria [12], eight out of 16 studies investigating FES in femoral fractures used these to diagnose FES [22,23,30,31,32,34,35,37]. Others relied on clinical evidence and a partial use of Gurd’s criteria [33]. Others used autopsies [26,29,36], ICD codes [46], or did not provide any information about their diagnostic approach [25,38,39].

For a better understanding of the variability observed, it is relevant to consider how many changes in the diagnostics have been made.

Following the first diagnostic approach by Gurd (1970), he defined the major and minor clinical criteria. This description continues to be the most widely used [12,47]. In 1983, Schonfeld presented a scoring system in which different symptoms of FES were rated. A patient scoring more than five points would be diagnosed with FES [9]. Lindeque et al. added a blood gas analysis and hypoxemia to the clinical signs in early FES [11]. Vedrinne et al. included pulmonary infiltrates, platelet counts, total blood lipids and the presence of long-bone fractures to taper the score for polytraumatized patients [10].

Of note, none of those diagnostic tools mentioned above were validated prospectively. This may be relevant to the inconsistency in the diagnostics of FES. It is assumed that, due to this diagnostic inconsistency, cases of FES have been missed in the past [19].

Another reason for misjudging the risk of FES is the lack of a description of the risk factors. While anatomical abnormalities, such as a patent foramen ovale [21], have been identified, and preoperative screening has been recommended, there is neither any medical prophylaxis for FES nor any distinct blood withdrawals, as can be seen for other major operations (e.g., an assessment of the coagulation status for joint replacements or heart surgery).

As far as the diagnostic criteria are concerned, it is evident that MRIs have increasingly been proposed [48] for the diagnosis of FES, including the cerebral form (CFES). In most recent publications, it has been used to exclude this complication in patients with acute FES. The presence of a “starfield” pattern, which demonstrates multiple punctate hyperintensities during T2 imaging, is considered a diagnostic for CFES [49].

With recent studies reporting the mortality rates of FES between 1% and 10% [22,24,25,26], this inconsistency in the diagnostics appears to call for a more standardized approach to the risk analysis and a rapid diagnosis. This appears to be relevant, since patients suffering from FES show rates of full recovery of approximately 90% when receiving proper supportive care [50].

Regarding our second question, there has been a significant decrease in the incidence of FES within the last decades. Numerous changes in trauma care over the last 60 years may play a role. Several studies reported cases of FES only in patients with delayed operative care [39,45]. Back in the 1950s and 1960s, the early fixation of fractures was believed to be a cause of FES [51,52], and the concept of patients being too sick to operate prevailed into the 1980s, leading to routinely delaying fracture fixation [53]. Subsequently, several clinical studies revealed that leaving long bone fractures unstabilized for several days dramatically increased the risk of extensive FES [45,54,55]. Therefore, several treatment principles changed towards the early stabilization of fractures, ideally within 24 to 72 h. All modern concepts of timing of surgical approaches, such as Early Total Care (ETC), Early Appropriate Care (EAC), Damage Control Surgery (DCS), and Safe Definitive Surgery (SDS), propose specific modifications in the surgical treatment in order to enable surgeons to perform fracture fixations as early as possible, some in a modified fashion [53,56,57,58].

For those patients that require an intramedullary fixation of long bone fractures, modifications of reamer systems were developed (slim reamer shafts, sharp reamer tips, and enlarged reamer flutes) that aimed to reduce the intramedullary pressure peaks during reaming, further reducing the embolic load. The effects of unreamed nailing were studied as well but did not appear to have certain effects on healing [59]. In 2005, the Reamer Irrigator Aspirator system (RIA) was developed that allows for the suction of the intramedullary contents with the intention to decrease the embolic load during reaming and to reduce the risk of intraoperative embolization [60,61]. Even though the clinical evidence of a significant effect of RIA on FES is still rare, in experimental studies, it proved to be effective [60,61,62]. It was demonstrated to be associated with a reduction in fat intravasations in a prospective clinical study [63].

Recently, the second generation was launched that allows for a better use in narrow canals and special situations [64], but its benefits are yet to be studied.

In summary, these changes may have contributed to further improvements in patient outcomes [57,65,66,67,68,69,70], and it appears to be justified to discuss their influence in the reduction of FES (Figure 1).

In terms of our third question, we found that the injury pattern has a significant impact on the rate of FES. In our data, the highest incidence of FES was found in bilateral acute femur fractures when compared with both unilateral traumatic and pathological fractures (Figure 2). This was in line with the fact that patients sustaining bilateral femoral fractures usually present with more concomitant injuries and a higher injury severity score (ISS). It must be kept in mind that, due to the structure of the ISS, the second femoral fracture (AIS 3 or 4) does not account for the total sum of ISS. Therefore, the severity of the impact may have attributed to the higher risk for FES in these patients and the associated soft tissue injuries [71].

These findings have been discussed in a similar fashion by other groups but certainly require special attention and further study [34,72].

The incidence of FES in patients with pathological fractures of the femur was also slightly but significantly higher compared to acute nonpathological unilateral fractures. This finding is not surprising, as the pathomechanisms of embolization in pathological fractures are accentuated by coagulopathy.

Therefore, among the theories of FES, the mechanical, the biomechanical, and the coagulation theories may be relevant for the development of FES in a variable fashion in our results. According to the mechanical theory [73], the intravasation of fat droplets occurs through acute soft tissue injury and intramedullary pressure changes. At the fracture site, this occurs when the intramedullary pressure exceeds the venous pressure and fat droplets are forced into the venous system. Those droplets are then collected in the pulmonary capillaries and can add to the mechanical obstruction, resulting in FES. Pell et al. proved that echogenic materials pass through the right atrium during reaming procedures [74] by transesophageal echocardiography. Although arteriovenous shunts are opened due to increased intrapulmonary pressures [75], further mechanisms appear to be relevant for the development of FES, as summarized in the coagulation theory. As thromboplastin is released, along with the fat from injured long bones, an acute activation of the clotting cascade occurs. Wenda et al. clearly showed an increase in the size of fat droplets during entrance into the lungs [76]. Histologically, the fat droplets were surrounded by platelet aggregations, thus further increasing the size of the initial fat droplet [76,77]. The procoagulant properties of the released fat are thought to be due to the presence of tissue factors that trigger thrombin generation and the subsequent formation of fibrin and activation of platelets, subsequently further increasing the size of the clot [78]. According to the biochemical theory, neutral fat droplets undergo degradation to free fatty acids (FFA), which can exert direct damage onto endothelial cells [79]. FFAs have been shown to cause damage to the capillary beds of the lungs [80]. The time required to produce these FFAs may explain the time delay seen in the onset of FES after trauma. Therefore, this mechanism may not be relevant for acute intraoperative embolization but plays a role in the subsequent development of FES. Along these lines, elevated C-reactive protein (CRP) levels may cause the circulating chylomicrons, the soluble form of triglycerides, to agglutinate into fat globules. This process may explain the presence of FE in the lungs of patients without trauma [81].

As far as recommendations towards surgical care are concerned, we feel that a number of improvements have added to the observed reduction of FES over time. Although Pinney described a slight trend of delays in surgery associated with higher incidences of FES, especially in younger men [35], their findings had no statistical significance. Of note, some studies also reported the onset of FES prior to surgery, even within 24 h after the initial trauma [19,82,83]. We hypothesize that the inclusion of endpoints of resuscitation [84], application of lactate clearance [85], and revised criteria for patients at special risk (borderline patients) have played a role [86]. We are aware that FES will occur in some patients irrespective of the timing of the definitive treatment [1].

In view of the pathogenetic differences between acute isolated fractures, pathological fractures, and the increased soft tissue trauma connected with bilateral fractures, our findings are not surprising. Nevertheless, it must be considered that FES, in some studies, may have been diagnosed in association with other causes of pulmonary failure, such as ARDS or ALI.

Regarding our fourth finding, despite extensive clinical and experimental research, there appears to be no universally accepted treatment recommendations for FES. The standard treatment still is supportive care aiming to maintain oxygenation and ventilation, supporting hemodynamics, and resuscitation with fluids and blood products [48,87,88,89,90]. Multiple attempts to establish targeted therapies, such as the application of heparin, aspirin, or steroids, have failed to provide beneficial effects [3,48,87]. Therefore, we cannot give any recommendations for the treatment or prevention of FES.

Our study has both strengths and limitations. On the one hand, to our knowledge, it is the first study that covers the entire literature since the clinical relevance of FES for orthopedic surgery was described. Furthermore, it reflects the various diagnostic criteria used, as listed in Table 1. Some studies reported about the incidence and clinical consequences of FES but did not indicate how it was diagnosed. Others used the International Classification of Diseases (ICD) code retrospectively or Gurd’s criteria with different levels of accuracy, while some made the diagnosis at autopsy. Finally, we included the pattern and cause of injury into our considerations. A clinical group looked at pathological fractures in detail before and suggested that a higher pulmonary embolic load can be expected during early intramedullary femoral fracture stabilization compared with primary external fixation [88,89]. They were also the first to report cognitive dysfunction associated with acute fat embolization during reamed nailing, describing the effects of reaming in isolated pathological fractures [90,91,92], and the effect of the injury pattern of the incidence of FES. Moreover, Hall et al. proved that the suction of intramedullary contents is associated with a less embolic load in terms of intraoperative embolization [63].

On the other hand, the limitations of our review are those that imply the natural issues associated with a systematic review, i.e., the quality of documentation of a given manuscript.

We also assessed the quality of studies when identical parameters were used to diagnose FES and found a wide variation in the level of accuracy. According to our assessment, the only study that revaluated Gurd’s criteria on a daily basis was published by Prakash et al., thus describing it as a continuous scoring system [23]. This study reported by far the highest rate of FES out of all the included studies, which might indicate that, with less strict application of the diagnostic criteria, a relevant number of FES cases might be missed.

This is in line with the previous experiences from our group. We previously hypothesized that the incidence of ARDS has decreased in patients with major trauma. However, while there appeared to be some evidence in the data from a trauma registry [2], we were not able to prove it in a systematic review [93]. In contrast, the results of the current systematic review point out that fat embolism appears to be undergoing a sustained reduction.

The great discrepancy in the reported incidences of FES may be due to its subtle clinical presentation, to inconsistencies in the diagnostics, or to other clinical changes. Moreover, the full-blown syndrome of fat embolism is less common, with incidences varying strongly from below 1% up to 23% in patients with long bone fractures [22,23]. Among the reasons for this is the fact that the pulmonary consequences of FES can be managed more easily by ventilatory support even though prolonged ICU stays may ensue [94,95,96].

Our current review did not provide any options to help decision-making to avoid fat embolization during the clinical course. In addition, although we included information on the diagnostic procedures in the given studies, we were unable to draw any meaningful conclusions in terms of recommendations on which diagnostic procedures to include. The inclusion of three studies that based the diagnosis of FES/FE on autopsies [26,29,36] may also have affected the results obtained. However, we feel that this reflects the clinical reality.

In summary, our research questions can be answered as follows. There is a strong inconsistency in the diagnostics of FES, with no universally accepted diagnostic tools. There has been a significant decrease in the incidence of FES after major fractures over the last decades. The injury pattern has a significant impact on the incidence of FES, with the incidence in bilateral femur fractures significantly higher than in unilateral fractures. There are no universally accepted evidence-based recommendations for the treatment or prevention of FES. We feel that, despite these positive trends, further research is justified to prevent FES in patients with acute injuries.

In conclusion, our study demonstrated that the diagnostic approach to FES is still highly inconsistent and, therefore, insufficient. There is no universally accepted diagnostic tool for FES, which probably has led to many cases of FES being missed. Due to this, patient safety can be jeopardized, causing preventable morbidity and mortality. Furthermore, the incidence of FES has significantly decreased over the last decades due to great changes in trauma care. When looking at the impact of injury patterns on the incidence of FES, we could clearly demonstrate that bilateral femur fractures increased the risk of developing FES compared to unilateral fractures. Finally, we must stress the fact that there was no universally accepted treatment recommendations or algorithms for fat embolism syndrome in the most recent literature.

## 5. Conclusions

In summary, our study demonstrated that the diagnostic approach to FES remains highly inconsistent. There has been a decrease in the incidence of FES after femur fractures over the last few decades, which we attribute to the improvements in ventilation and general resuscitation algorithms, as well as to changes in the timing of definitive surgical treatment. The type of injury pattern appears to play a role regarding the risk of FES, with the incidence in bilateral femur fractures being significantly higher than in unilateral fractures. There are no targeted evidence-based recommendations for the treatment or prevention of FES. Concerning the diagnostic criteria, it is evident that MRIs have increasingly been proposed [48] for the diagnosis of FES, including the cerebral form. We feel that, despite these positive trends, further study is justified to help reduce FES in patients with acute injuries.

## Figures and Tables

**Figure 1 jcm-10-02733-f001:**
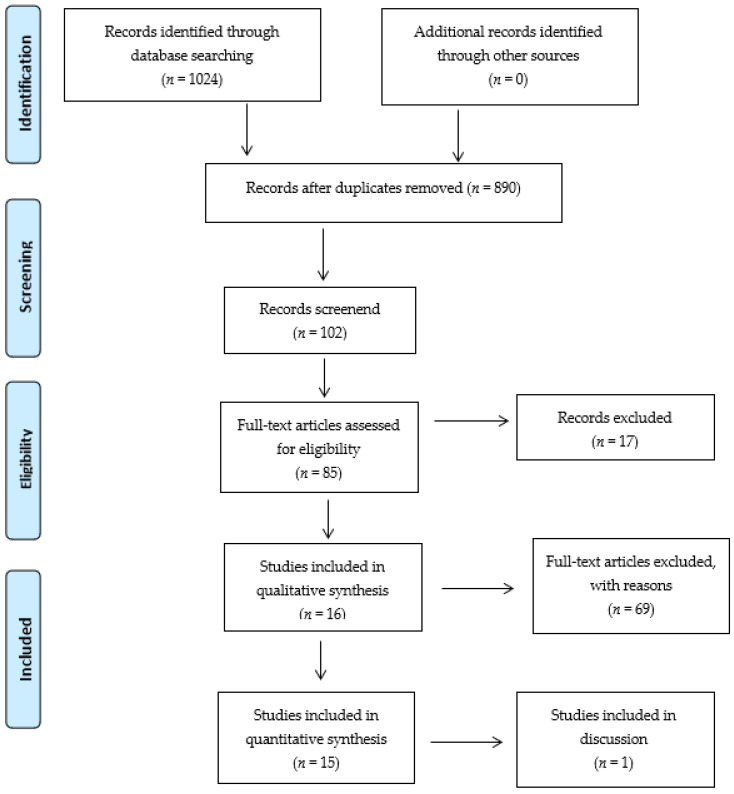
Preferred Reporting Items for Systematic Reviews and Meta-analyses (PRISMA) flow diagram of the study selection.

**Figure 2 jcm-10-02733-f002:**
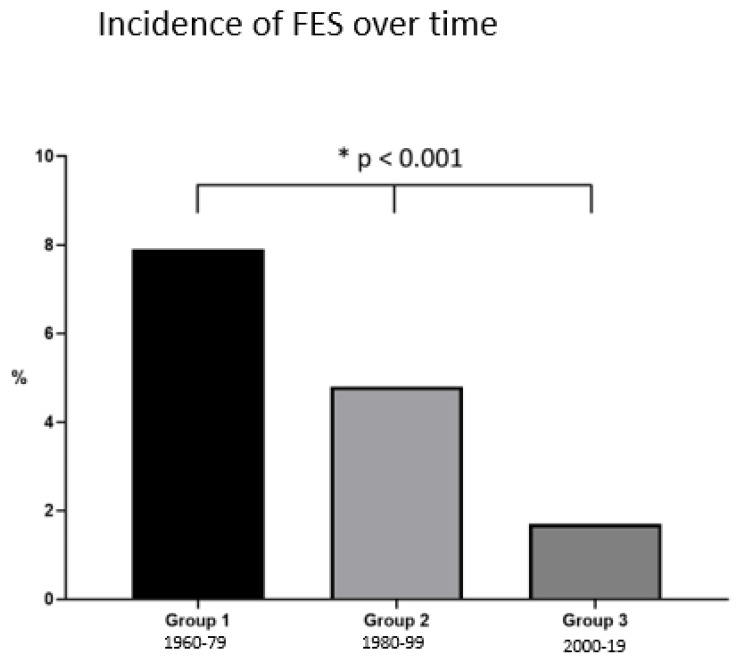
Incidence of clinically diagnosed FES over time (level of significance gr1 versus gr2 and gr2 versus gr3).

**Figure 3 jcm-10-02733-f003:**
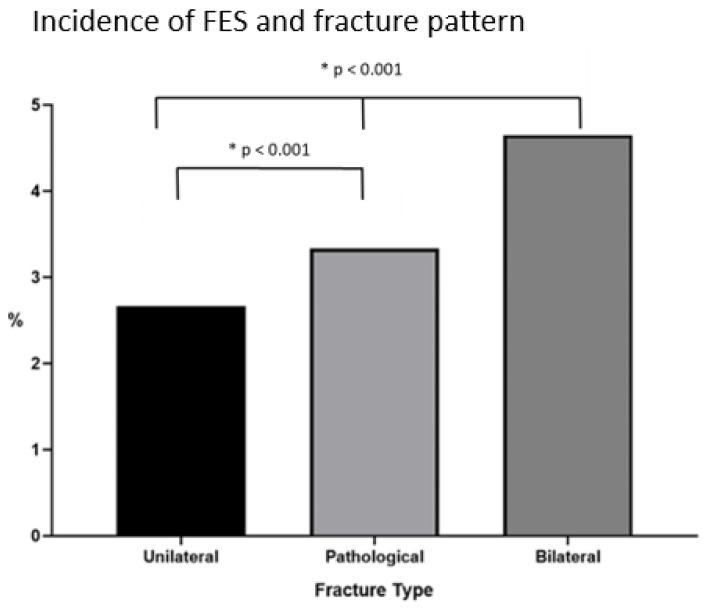
Association between the injury pattern of long bone fractures (femur) and the incidence of FES.

**Table 1 jcm-10-02733-t001:** Risk of bias for nonrandomized trials.

Study	Confounding	Subject Selection	Intervention Measurement	Deviation from Intended Intervention	Attrition	Outcomes Measurement	Selective Reporting	Overall
Cole et al. [23]	Low	Low	Low	Moderate	Low	Moderate	Serious	Serious
Prakash et al. [25]	Moderate	Low	Low	Low	Low	Moderate	Moderate	Moderate
Fabian et al. [29]	Moderate	Moderate	Low	Moderate	Low	Low	Moderate	Moderate
Talucci et al. [30]	Moderate	Critical	Moderate	Moderate	Low	Moderate	Serious	Critical
Ten Duis et al. [31]	Moderate	Moderate	Low	Serious	Low	Moderate	Moderate	Serious

**Table 2 jcm-10-02733-t002:** Study populations.

Author	Study Period	Total Patients (n)	Injury Pattern	Trauma Mechanism	Fracture	Sex (M/F)	Age Range (Years)
Ten Duis [22] *	1967–1985	172	U	blunt	21 open	133/29 *	16–65
					151 closed		
Talucci [23]	1978–1981	100	U	n.a.	n.a.	n.a.	n.a.
Veith [25]	1968–1978	54	U	blunt	open/	32/22	15–58
					closed		
King [26]	1972–1977	112	U	blunt	19 open	n.a.	n.a.
					93 closed		
Bone [29]	1985–1987	79	U	n.a.	n.a.	n.a.	n.a.
Bonneviale [30]	1986–1999	40	Bi	blunt	n.a.	27/13	17–50
Pinney [31]	1987–1994	274	U	n.a.	n.a.	189/85	13–96
Cole [32]	1990–1996	65	Pa	n.a.	n.a.	n.a.	15–92
Fabian [33]	1990	92	U	n.a.	n.a.	n.a.	n.a.
Assal [34]	1994–1997	10	Pa	n.a.	n.a.	3/7	53–99
Barwood [35]	1994–1997	43	Pa	n.a.	closed	17/26	39–93
Tsai [36]	1997–2008	1541	U	n.a.	n.a.	n.a.	n.a.
Cannada [37]	2000–2004	89	Bi	blunt	n.a.	46/43	16–63
Prakash [38]	2010–2011	48	U	n.a.	closed	38/10	16–40
Silva [39]	2011–2015	272	U	n.a.	n.a.	229/43	>16

Notes: * Miscalculation in the published paper. Our statistics are based on the total number of patients (172). U = unilateral; Bi = bilateral; Pa = pathological; n.a. = not available.

**Table 3 jcm-10-02733-t003:** Diagnostic tools/criteria applied to diagnose fat embolism syndrome (FES).

Author	Period	Incidence of FES (%)	Diagnostic Criteria for FES
Ten Duis et al. [22]	1967–1985	3.5	GURD
Talucci et al. [23]	1978–1981	5	GURD
Veith et al. [25]	1968–1978	13	1 out of GURD
King et al. [26]	1972–1977	15.18	“clinical evidence”
Bone et al. [29]	1985–1987	1.12	n.a.
Bonneviale et al. [30]	1986–1999	5	GURD
Pinney et al. [31]	1987–1994	4	GURD
Cole et al. [32]	1990–1996	3.08	Autopsy
Fabian et al. [33]	to1990 start unknown	10.87	n.a.
Assal et al. [34]	1994–1997	10	Autopsy
Barwood et al. [35]	1994–1997	2.2	CRVD
Tsai et al. [36]	1997–2008	0.78	GURD
Cannada et al. [37]	2000–2004	4	GURD
Prakash et al. [38]	2010–2011	23	GURD
Silva et al. [39]	2011–2015	4	n.a.

Notes: GURD = Gurd’s guidelines; CRVD = acute cardiorespiratory and vascular dysfunction; ICD = International Classification of Diseases; n.a. = not available.

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
