# Peer review of "Incidence of Fat Embolism Syndrome in Femur Fractures and Its Associated Risk Factors over Time—A Systematic Review"

_jcm, 2021, doi:10.3390/jcm10122733_

Round 1
Reviewer 1 Report
Excellent manuscript on in important topic. I have minor suggestions.
The date format for collection periods in abstract is difficult to read. Recommend presenting like you did in methods section (Jan 1, 1960 – Dec 31, 1969)
May be helpful to include gurd’s criteria. Most readers will be unfamiliar with this.
In your conclusion you discuss diagnostic techniques such as MRI and cerebral form. This warrants further details in the discussion. Adding a paragraph on this may be helpful to orient readers as I am unfamiliar.
In the conclusion should you mention the main hypothesis of why FES has decreased? Better resuscitation, endpoints, and ventilator care.
Author Response
- The date format for collection periods in abstract is difficult to read. Recommend presenting like you did in methods section (Jan 1, 1960 – Dec 31, 1969) - thank you for your comment, we made changes accordinlgy.
- May be helpful to include gurd’s criteria. Most readers will be unfamiliar with this. - Thank you for this precious comment, we added the Gurd´s criteria to our manuscript.
- In your conclusion you discuss diagnostic techniques such as MRI and cerebral form. This warrants further details in the discussion. Adding a paragraph on this may be helpful to orient readers as I am unfamiliar. - Thank you for your comment - we have made changes accordingly.
- In the conclusion should you mention the main hypothesis of why FES has decreased? Better resuscitation, endpoints, and ventilator care. - Thank you for your comment, we totally agree with you and made changes accordingly.
Reviewer 2 Report
The authors conducted a literature and review over a large span of time 1960-2019. Two independent reviewers conducted data extraction and analysis to examine the field of fat embolism and examine clinical associated factors.The search terms only covered until 2019. It is 2021.
The authors found a high association of the traumatic high-energy bilateral femur fractures and fat embolism. This would have been expected.
The authors also found that the issue of fat embolism has decreased in impact over time.
Author Response
- The search terms only covered until 2019. It is 2021 - There has been some delay between finishing the literature research and the final publication due to the ongoing pandemic. Nevertheless, there have been no new publications in that period of time, that would have statistically impacted this review.
- The authors found a high association of the traumatic high-energy bilateral femur fractures and fat embolism. This would have been expected. - Thank you for your comment.
- The authors also found that the issue of fat embolism has decreased in impact over time - Thank you for your comment.
Reviewer 3 Report
This is an interesting review article about the incidence and the risk factors for fat embolism following fractures of the femur. This topic is quite interesting and the questions that the authors try to answer with this paper are important for the orthopaedic community. Even though the topic is interesting and the authors have made a significant effort to produce this paper, the presentation needs some changes in order to enhance the manuscript and make it better for the reader. Following I present my comments in a point by point manner.
Title: The phase 'comparative analysis' is not so relevant for this manuscript, consider removing it.
Abstract: The content of the abstract is good, but a short conclusion is missing from it. Also some ':' are missing after each subtopic. Consider changing the subtitle 'statistics' to 'results' and combining all the subtitles in the Methods section to one wider one, such as 'materials and methods'.
Introduction: Consider rephrasing the phrase 'fully-blown syndrome', this is not an expression that is used regularly. The introduction is too short as well. The authors try in only 10 lines make a historical review and explain why this review is needed and present their hypothesis as well. Please consider fragmenting there introduction to two-thee paragraphs and explain why your questions are relevant to the orthopaedic surgeons.
Methods:
In the search strategy section what is the meaning of 'relevant combinations of these terms'? Please clarify also the second way of identifying relevant studies, which is not clear at the moment (did the author's checked the references inside some articles to see if they have missed any? and did they finally identify any new articles through that technique? Please comment why these have been missed in the first search as well, maybe the search criteria were not wide / inclusive enough - how do we know that there are no other studies missed?)
In the inclusion/exclusion criteria please specify why you excluded case reports from your study.
In the data extraction and analysis section the presentation of the data collection is really confusing, consider combine them in a single text format and not as bullet points. Also it looks like that the authors considered different pattern id the fracture was unilateral, bilateral or pathological. It is unclear if they have included all the femoral fractures (neck of femur, subtrochanteric, shaft or distal femoral fractures) or only the femoral shaft fractures. Was the energy (high or low) of trauma a factor you have checked as well? At the end of the section the phrase 'diagnostic criteria' is referring to the fracture of the fat embolism. Also the time to surgical treatment and the mortality are unclear in your results. Also consider changing the term 'fracture pattern' as this term is widely used for other reasons to describe if the fracture is spiral/oblique etc.
The definitions section is not relevant in this part of the study, consider removing it.
The authors divided the groups every 20 years, please consider explaining why in this section and not defer to the discussion section. Were the time frames randomly chosen? Was there a reason of the specific cut-offs? Was there a treatment change or diagnostic change during these periods?
In the statistics section (which consider changing to 'statistical analysis' section), the only statistical test the authors used was the x2. Please consider double checking if this is the only and the most appropriate statistical test required.
The Figure one is not a full PRISMA flowchart. Each step has to show the number of articles excluded and why. In one of the steps the authors mention 'excluded with reasons' - what are the reasons?
Results: The text in most of this section is referring to the tables without giving the most important results. This is quite confusing for the reader. Consider adding some relevant and important information on the actual text and leave the rest of the data on the tables.
Discussion: The whole section has to be re-written. The article has to be written in passive voice (avoid using the phrases 'we' and 'I'). At the moment the authors start the discussion with the limitations of the study and it feels like they thy to defend their choices. The limitations and the arguments against them extend to the half of the discussion. The discussion has to start with the most important findings of the review, analyse point by point the significant results and present discussion from the literature. At the moment the answer for the first question extends only for 10 lines and uses only a small amount of references. Also there is a mention about the RIA reaming technique. RIA is not widely used over the world routinely and it is unclear how it is relevant for the present study.
Regarding answer to the third question it is clear that there are no universally accepted recommendations, maybe this manuscript using all this data may conclude to a recommendation. Is this possible?
Hope that these comments help to enhance your manuscript.
Kind regards,
Author Response
Title: The phase 'comparative analysis' is not so relevant for this manuscript, consider removing it. – Thank you for your comment, we agree with you and made changes accordingly.
Abstract: The content of the abstract is good, but a short conclusion is missing from it. Also some ':' are missing after each subtopic. Consider changing the subtitle 'statistics' to 'results' and combining all the subtitles in the Methods section to one wider one, such as 'materials and methods'. – Thank you for your precious comment. We made changes accordingly.
Introduction: Consider rephrasing the phrase 'fully-blown syndrome', this is not an expression that is used regularly. The introduction is too short as well. The authors try in only 10 lines make a historical review and explain why this review is needed and present their hypothesis as well. Please consider fragmenting there introduction to two-thee paragraphs and explain why your questions are relevant to the orthopaedic surgeons. – Thank you for your precious comment. We made changes accordingly.
Methods:
In the search strategy section what is the meaning of 'relevant combinations of these terms'? – this statement was deleted; it did not make sense. Thanks for your comment.
Please clarify also the second way of identifying relevant studies, which is not clear at the moment (did the author's checked the references inside some articles to see if they have missed any? and did they finally identify any new articles through that technique? Please comment why these have been missed in the first search as well, maybe the search criteria were not wide / inclusive enough - how do we know that there are no other studies missed?) – in our opinion that is a common way of performing a thorough literature research as some publications might not be available online. In such cases our library would organize those publication for us so we could also screen them. Fabian et al. for example was such a case.
In the inclusion/exclusion criteria please specify why you excluded case reports from your study. That was a decision, we made prior to the literature research when establishing our search protocol. This decision might be discussed or challenged.
In the data extraction and analysis section the presentation of the data collection is really confusing, consider combine them in a single text format and not as bullet points. – Changes were made accordingly.
Also it looks like that the authors considered different pattern id the fracture was unilateral, bilateral or pathological. It is unclear if they have included all the femoral fractures (neck of femur, subtrochanteric, shaft or distal femoral fractures) or only the femoral shaft fractures. We included all types of femoral fracture in our search, the included papers only report on shaft fractures or don’t give specific information on the exact type of fractures though.
Was the energy (high or low) of trauma a factor you have checked as well? – This would have been very interesting but the information in the publications was too scarce.
At the end of the section the phrase 'diagnostic criteria' is referring to the fracture of the fat embolism. - Here we refer to the diagnostic criteria of FES.
Also the time to surgical treatment and the mortality are unclear in your results. – This would have been very interesting variables to look at, but the literature did not give sufficient information.
Also consider changing the term 'fracture pattern' as this term is widely used for other reasons to describe if the fracture is spiral/oblique etc. – Very helpful comment, we changed it to injury pattern.
The definitions section is not relevant in this part of the study, consider removing it. – We agree, it was removed. Thank you for the comment.
The authors divided the groups every 20 years, please consider explaining why in this section and not defer to the discussion section. Were the time frames randomly chosen? Was there a reason of the specific cut-offs? Was there a treatment change or diagnostic change during these periods? – The time frames were not chosen completely randomly. In the first 2 decades very few changes in the treatment of long bone fractures or ventilation were promoted. Later on, ventilation techniques drastically improved. Finally in the last two decades timing of definitive fracture fixation became more respectful towards the overall patients ‘condition. Therefore we think that these time frames “vaguely” represent major changes in the treatment of orthopedic trauma. We do not believe that in-depth explanation belongs in the Methods section and would therefore refer to making this explanation in this section. Your comment though was very much appreciated.
In the statistics section (which considers changing to 'statistical analysis' section), the only statistical test the authors used was the x2. Please consider double checking if this is the only and the most appropriate statistical test required. - We thank the reviewer for the thorough review of the manuscript. Following your recommendations, we have re-checked the statistics and improved the wording accordingly.
The Figure one is not a full PRISMA flowchart. Each step has to show the number of articles excluded and why. In one of the steps the authors mention 'excluded with reasons' - what are the reasons? Those numbers were included in the Figure we submitted but apparently got lost during formatting, we have resubmitted the original containing the numbers. The reasons for exclusion are mentioned in the “Inclusion and Exclusion Criteria” section.
Results: The text in most of this section is referring to the tables without giving the most important results. This is quite confusing for the reader. Consider adding some relevant and important information on the actual text and leave the rest of the data on the tables. Thank you for your comment and thorough study of our manuscript. We did not feel though, that changes are needed in this section, as the objective of the result section is to only deliver the most important facts.
Discussion: The whole section has to be re-written. The article has to be written in passive voice (avoid using the phrases 'we' and 'I'). At the moment the authors start the discussion with the limitations of the study and it feels like they thy to defend their choices. The limitations and the arguments against them extend to the half of the discussion. The discussion has to start with the most important findings of the review, analyse point by point the significant results and present discussion from the literature. At the moment the answer for the first question extends only for 10 lines and uses only a small amount of references. Also there is a mention about the RIA reaming technique. RIA is not widely used over the world routinely and it is unclear how it is relevant for the present study. – Thank you for your comment, we did rewrite the whole section. We do use RIA on a regular basis. Therfore we explained in deeper detail now why we think it is relevant.
Regarding answer to the third question it is clear that there are no universally accepted recommendations, maybe this manuscript using all this data may conclude to a recommendation. Is this possible? This is unfortunately not possible, as the date given by the literature is extremely heterogenic.
Round 2
Reviewer 2 Report
The authors addressed my comment regarding the gap in time between the end of the literature search and the review of this manuscript. This response is satisfactory.